# Anytime Verified Agents: Adaptive Compute Allocation for Reliable LLM Reasoning under Budget Constraints

**Dipkumar Patel**  *dip@llmsresearch.com*
*LLMs Research Inc.*

**Reviewed on OpenReview:** *https://openreview.net/forum?id=JMDCMf7mlF*

## Abstract

Large language model (LLM) agents can perform multi-step reasoning, planning, and tool use. However, their performance scales with the computational budget. Existing methods allocate computational resources using static strategies such as fixed search depths, constant self-consistency sampling, or uniform verification, so simple problems can consume as much compute as complex tasks. We present Anytime Verified Agents (AVA), a framework that dynamically allocates compute across search, sampling, and verification within a user-specified budget, with an extensible interface for tool use. AVA combines calibrated uncertainty estimation, value-of-information-guided search expansion, and selective verification cascades with early exits. The controller allocates compute based on uncertainty and estimated marginal reliability gains. AVA is evaluated on mathematical reasoning (GSM8K and MATH), multi-hop question answering (HotpotQA), and code generation (HumanEval), with two model backends (GPT-5 and GPT-4o), and compared to fixed-depth search, self-consistency, and always-verify baselines. Across these benchmarks, AVA reduces cost at matched reliability thresholds while maintaining comparable accuracy.

## 1 Introduction

Agentic large language models (LLMs) can plan, reason, and interact with external tools to solve complex tasks (Wei et al., 2022; Yao et al., 2023a; Zhou et al., 2023). Recent work has explored deliberate reasoning, external validation, and adaptive computation to improve reliability under test-time constraints.

Deploying these agents in real-world applications involves a trade-off: improving reliability through increased test-time computation comes with increased cost. Current approaches address reliability through techniques like self-consistency (Wang et al., 2023), tree search (Yao et al., 2023a; Besta et al., 2024), and external verification (Cobbe et al., 2021), yet they allocate compute resources statically using fixed search depths, constant sampling budgets, or uniform verification strategies regardless of problem difficulty.

Static allocation can be inefficient. Consider a mathematical reasoning agent: a simple arithmetic problem may require only a single forward pass, while a multi-step proof demands extensive search and verification. Applying the same computational budget to both wastes resources on easy problems and under-allocates to hard ones, limiting reliability under a fixed budget. Moreover, static strategies cannot adapt to varying uncertainty levels or changing marginal returns on additional compute.

Recent work has explored adaptive computation (Kreutzer et al., 2018; Ahn et al., 2024) and selective verification, typically focusing on individual compute dimensions. To the best of our knowledge, prior work has not jointly allocated compute across search, sampling, verification, and tool use under unified budget constraints based on task difficulty, uncertainty estimates, and expected marginal gains.

This paper introduces **Anytime Verified Agents (AVA)**, a framework that adaptively allocates computational resources to maximize reliability under user-specified budgets. AVA makes three key contributions:

1. **Budget-aware controller**: A decision-making module that dynamically allocates tokens and verification passes based on calibrated uncertainty estimates and predicted marginal reliability gains per unit cost. The controller additionally supports tool-call allocation, though this dimension is not exercised in the current evaluation.

2. **Selective verification cascade**: A multi-tier verification system with early exits that routes problems through progressively stronger (and costlier) verifiers only when uncertainty exceeds calibrated thresholds.

3. **Value-of-information guided search**: An adaptive tree search mechanism that expands nodes only when the expected information gain exceeds the marginal cost, allowing efficient exploration under budget constraints.

We evaluate AVA on mathematical reasoning (GSM8K (Cobbe et al., 2021) and MATH (Hendrycks et al., 2021)), multi-hop question answering (HotpotQA (Yang et al., 2018)), and code generation (HumanEval (Chen et al., 2021)), comparing against fixed-depth tree search, self-consistency, and always-verify baselines across multiple budget levels and two model backends (GPT-5 and GPT-4o). Across these benchmarks, AVA reduces cost at equivalent reliability thresholds while maintaining comparable accuracy.

## 2 Related Work

Our work synthesizes advances across agentic reasoning, adaptive computation, verification strategies, and budget-constrained reliability.

**Reasoning and Search.** Chain-of-Thought (CoT) prompting (Wei et al., 2022) demonstrated that explicit step-by-step reasoning improves LLM performance and remains single-pass. Tree-of-Thoughts (ToT) (Yao et al., 2023a) explores multiple reasoning paths via search trees; common implementations use fixed branching factors and depths regardless of problem difficulty. Language Agent Tree Search (LATS) (Zhou et al., 2024) integrates MCTS with uncertainty-guided expansion, yet maintains fixed total search budgets. Graph of Thoughts (Besta et al., 2024) generalizes to arbitrary graphs and retains static allocation. Least-to-Most (Zhou et al., 2023) decomposes problems sequentially using fixed strategies.

**Sampling and Verification.** Self-consistency (Wang et al., 2023) samples multiple completions for majority voting and uses fixed sample counts across problems. Best-of-N and AlphaLLM (Tian et al., 2024) improve efficiency while maintaining static policies. Training verifiers (Cobbe et al., 2021) scores reasoning chains and verifies all candidates uniformly. Cascaded verification systems reduce costs via early exits and use fixed routing policies. PAL (Gao et al., 2023) uses execution for validation without adaptive allocation.

**Adaptive Computation.** Early stopping (Prechelt, 1998) and confidence-based halting (Kreutzer et al., 2018) adapt per-instance and operate on single forward passes. D-LLM (Jiang et al., 2024) adapts per-token compute allocation within transformer layers but focuses on internal model efficiency rather than unified budget allocation across reasoning steps. Tool use frameworks (Toolformer (Schick et al., 2023), ReAct (Yao et al., 2023b), AnyTool (Du et al., 2024)) and self-reflection methods (Reflexion (Shinn et al., 2023), Self-Refine (Madaan et al., 2023)) improve capabilities while using fixed budgets and iteration limits.

**Uncertainty Estimation.** Raw LLM log-probabilities are poorly calibrated (Guo et al., 2017), requiring post-hoc calibration (Zadrozny & Elkan, 2001). Self-consistency signals provide confidence proxies but don't capture multi-step trajectory uncertainty. Existing methods operate on per-instance predictions rather than trajectory-level uncertainty across search, sampling, and verification.

**Adaptive Token and Sample Allocation.** Several recent methods tackle compute allocation for LLM reasoning. SelfBudgeter (Li et al., 2025) trains models through reinforcement learning to estimate token budgets before generating a response, compressing outputs by roughly 60% on math benchmarks. The approach works well but adapts only token count and requires fine-tuning. Strategic Scaling (Zuo & Zhu, 2025) takes a different tack: it treats sample allocation as a bandit problem, learning on-the-fly which queries benefit from extra samples. This sidesteps training costs but only controls sampling, not verification or search.

Table 1: Comparison of compute allocation strategies. AVA adaptively allocates across all dimensions under unified budget constraints. *Tool-call allocation is supported but not evaluated in the current experiments.

| Approach | Search | Sampling | Verification | Tool Use | Budget |
|---|---|---|---|---|---|
| Self-Consistency (Wang et al., 2023) | Fixed | Adap. (fixed N) | None | None | Static |
| ToT (Yao et al., 2023a) | Adap. (fixed bud.) | Fixed | None | None | Static |
| LATS (Zhou et al., 2024) | Adap. (UCB) | Fixed | None | Fixed | Static |
| Verif. Cascades | Fixed | Fixed | Adap. (routing) | None | Partial |
| AnyTool (Du et al., 2024) | Fixed | Fixed | None | Adap. (routing) | Partial |
| D-LLM (Jiang et al., 2024) | Adap. (per-token) | Fixed | None | None | Partial |
| SelfBudgeter (Li et al., 2025) | Fixed | Adap. (learned) | None | None | Single |
| Strat. Scaling (Zuo & Zhu, 2025) | Fixed | Adap. (bandit) | None | None | Single |
| **AVA (Ours)** | **Adaptive** | **Adaptive** | **Adaptive** | Adaptive* | **Unified** |

AVA differs from these approaches in that it jointly controls search depth, sampling, verification intensity, and tool use under a unified budget. It relies on calibrated uncertainty from multiple signals instead of a learned predictor or bandit exploration, and integrates cascaded verification with early exits. These ideas are complementary: learned budget prediction could inform our controller, and online learning could be used to adapt thresholds during deployment.

Table 1 compares prior methods along the compute-allocation dimensions they control. Current methods fall into three categories: (1) static allocation (fixed search depth in ToT/LATS, constant sampling in self-consistency, uniform verification), (2) partial adaptation (adaptive search but fixed verification, or selective verification but fixed search), or (3) cost optimization without reliability guarantees.

# 3 Method

AVA adaptively allocates compute across search, sampling, verification, and tool use to maximize reliability under budget constraints. This section presents the framework, organized around three stages: uncertainty estimation (Section 3.2), budget-aware allocation (Section 3.3), and execution (Section 3.4). Figure 2 (Appendix B) provides the full architectural diagram.

## 3.1 Overview and Notation

Figure 2 (Appendix B) illustrates the AVA framework architecture. Given an input prompt $x$ and compute budget $B = (B_{\text{tok}}, B_{\text{tool}}, B_{\text{ver}})$, AVA operates through an iterative decision loop with three core stages: uncertainty estimation, budget-aware allocation, and execution.

To illustrate, consider a single AVA iteration. Stage 1 aggregates whichever confidence signals are available and calibrates them to produce $c \in [0, 1]$. Stage 2 computes the confidence gap $\Delta = r_{\text{target}} - c$ and applies the allocation policy (Algorithm 1) to determine resources for sampling, search, verification, and tool use. Stage 3 executes these actions, updates the state, and returns to Stage 1 until $c \geq r_{\text{target}}$ or budget $B$ is exhausted.

Table 2 summarizes the notation used throughout this paper.

**Stage 1: Uncertainty Estimation** (Section 3.2). AVA defines four confidence signals: token-level entropy ($c_H$), self-consistency ($c_S$), verifier scores ($c_V$), and trajectory features ($c_T$). Whichever signals are available at a given iteration are combined into a calibrated confidence $c$ via weighted aggregation (with renormalized weights) and isotonic regression.

**Stage 2: Budget-Aware Controller** (Section 3.3). Given state $\mathbf{s}_t$, the controller computes the confidence gap $\Delta = r_{\text{target}} - c$ and determines allocation across four compute dimensions (sampling, search, verification, tool calls) using a rule-based policy that prioritizes high expected reliability gain per unit cost.

**Stage 3: Execution** (Section 3.4). AVA executes the allocated actions: generates samples for self-consistency voting, runs value-of-information guided tree search, applies a verification cascade with early exits, and calls tools if allocated. After execution, the loop returns to Stage 1 for re-estimation.

Table 2: Notation used throughout the paper.

| Symbol | Definition |
| --- | --- |
| $x, y$ | Input prompt, generated output |
| $B = (B_{\text{tok}}, B_{\text{tool}}, B_{\text{ver}})$ | Compute budget: tokens, tool calls, verification passes |
| $B_r \in [0, 1]$ | Remaining budget fraction |
| $c \in [0, 1]$ | Calibrated confidence |
| $c_{\text{raw}}$ | Raw (uncalibrated) confidence |
| $c_H, c_S, c_V, c_T$ | Confidence from entropy, self-consistency, verifier, trajectory |
| $w_H, w_S, w_V, w_T$ | Signal weights (default: 0.3, 0.4, 0.2, 0.1) |
| $u = 1 - c$ | Uncertainty |
| $r_{\text{target}}$ | Target reliability threshold (default: 0.9) |
| $\Delta = r_{\text{target}} - c$ | Confidence gap |
| $t \in [0, 1]$ | Task complexity (from input features) |
| $d, n$ | Search depth reached, nodes expanded |
| $k$ | Number of samples for self-consistency voting |
| $b(u)$ | Branching factor as a function of uncertainty (Eq. 10) |
| $L$ | Number of verification cascade levels (default: 3) |
| $d_{\text{max}}$ | Cumulative search depth normalization constant (default: 10) |
| $h_t$ | Decision history up to step $t$ |
| $\mathbf{s}_t$ | Controller state: $(c_t, B_{r,t}, d_t, n_t, t_t, h_t)$ |
| $f^*$ | Calibration function (isotonic regression via PAVA) |
| $V_l$ | Level-$l$ verifier, returning $(v_l, s_l)$: validity and confidence in verdict |
| $\lambda$ | Depth decay parameter for VoI search (default: 0.2) |

**Example**. For "What is $17 \times 23$?" with budget 600 tokens and $r_{\text{target}} = 0.9$, initial confidence $c = 0.72$ yields $\Delta = 0.18$. The controller (Algorithm 1) allocates: $k = 5$ samples ($\approx$250 tokens), depth-1 search with breadth 2 ($\approx$100 tokens), and level-1 verifier ($\approx$50 tokens). Samples agree and the verifier confirms, raising confidence to 0.89 and approaching the target with minimal further compute.

## 3.2 Stage 1: Uncertainty Estimation and Calibration

Reliable uncertainty estimation is a prerequisite for adaptive allocation. Raw LLM outputs provide multiple uncertainty signals, but these are uncalibrated and must be aggregated.

### 3.2.1 Uncertainty Signals

AVA combines four complementary signals into a unified confidence estimate.

**Token-level entropy**. For generation $y$ with tokens $(t_1, \dots, t_n)$, we compute Shannon entropy from model log-probabilities:

$$H_{\text{token}}(y) = -\sum_{i=1}^{n} p(t_i \mid y_{<i}, x) \log p(t_i \mid y_{<i}, x). \tag{1}$$

Higher entropy indicates greater uncertainty. We convert to a confidence signal:

$$c_H = 1 - \min(1, \ H_{\text{token}}/\tau_H), \tag{2}$$

where $\tau_H$ is a normalization constant chosen on validation data so that most per-sequence entropy values fall below $\tau_H$ (default $\tau_H = 10$; see Appendix A for all threshold values and tuning methodology).

**Self-consistency signal**. For $N$ samples with answer set $\mathcal{A}$, we compute the vote entropy and normalize:

$$H_{\text{cons}} = -\sum_{a \in \mathcal{A}} \frac{\text{count}(a)}{N} \log \frac{\text{count}(a)}{N}, \qquad \tilde{H}_{\text{cons}} = \frac{H_{\text{cons}}}{\log(|\mathcal{A}|)}. \tag{3}$$

Confidence is $c_S = 1 - \tilde{H}_{\text{cons}}$, with $c_S = 1$ by convention when $|\mathcal{A}| = 1$ (all samples agree). Uniform disagreement gives $c_S \approx 0$.

**Verifier scores**. External verifiers (defined formally in Section 3.4.2) return scores $s_v \in [0, 1]$ indicating validity. For cascaded verifiers, we take the maximum score across levels reached: $c_V = \max_{l \in \{1, \dots, L\}} s_l$.

**Trajectory features**. Search depth $d$ and remaining budget fraction $B_r$ provide trajectory-based uncertainty signals:

$$H_{\text{traj}}(d, B_r) = 0.6\,(1 - d/d_{\max}) + 0.4\,(1 - B_r), \tag{4}$$

where $d_{\max} = 10$ is a normalization constant representing the maximum cumulative search depth across all iterations of the AVA loop (distinct from the per-iteration depth cap of 5 in Algorithm 1, which limits a single allocation step). Confidence: $c_T = 1 - H_{\text{traj}}$. Low search depth relative to $d_{\max}$ and low remaining budget both indicate higher uncertainty.

### 3.2.2 Aggregation and Calibration

Signals are combined via weighted average with dataset-specific weights learned via grid search on validation data:

$$c_{\text{raw}} = w_H \cdot c_H + w_S \cdot c_S + w_V \cdot c_V + w_T \cdot c_T, \tag{5}$$

with default weights $(w_H, w_S, w_V, w_T) = (0.3, 0.4, 0.2, 0.1)$ summing to 1.0. When a signal is unavailable, its weight is dropped and the remaining weights are renormalized. In the current experiments, $c_t$ at each iteration is computed from the two signals available before execution: token entropy ($c_H$) and trajectory features ($c_T$), aggregated via Equation 5 with renormalized weights $w_H/(w_H + w_T) = 0.75$ and $w_T/(w_H + w_T) = 0.25$. Self-consistency agreement and verifier confidence are not fed back into $c_t$ through Equation 5; instead, they influence decisions through their respective mechanisms (majority voting for answer selection, cascade confidence for termination). The aggregation framework supports all four signals; routing self-consistency and verifier feedback into the calibrated estimate is a straightforward extension that would change only the weight renormalization at each iteration.

**Post-hoc calibration**. Raw confidence estimates from neural networks are poorly calibrated (Guo et al., 2017). We apply isotonic regression on held-out validation data, fitting a monotonic function $f^* : [0, 1] \to [0, 1]$ via the pool adjacent violators algorithm (PAVA) (Barlow et al., 1972). Specifically, given $N$ validation examples with predicted confidences $\{p_i\}$ and correctness labels $\{y_i \in \{0, 1\}\}$, we solve

$$f^* = \arg\min_{f \text{ monotone}} \sum_{i=1}^{N} \big(f(p_i) - y_i\big)^2. \tag{6}$$

The calibrated confidence is $c = f^*(c_{\text{raw}})$, ensuring that reported confidence approximates the true success probability. Calibration is performed separately per dataset to account for domain-specific confidence distributions. Expected Calibration Error (ECE) on the calibration set is checked to validate quality (target ECE $< 0.05$).

## 3.3 Stage 2: Budget-Aware Controller

Given the uncertainty estimate from Stage 1, the controller decides how to allocate compute across four dimensions.

### 3.3.1 State Representation and Confidence Gap

At step $t$, the controller receives state $\mathbf{s}_t = (c_t, B_{r,t}, d_t, n_t, t_t, h_t)$, whose components are: calibrated confidence $c_t \in [0, 1]$, remaining budget fraction $B_{r,t} \in [0, 1]$, search depth reached $d_t$, and nodes expanded $n_t$. Task complexity $t_t \in [0, 1]$ is a normalized weighted sum of input length (token count / 512), parse depth (AST depth / 10, or 0 when no parse tree is available), and a domain indicator (1 for code, 0.5 for math, 0 for QA), with equal weights and clipping to $[0, 1]$. Decision history $h_t$ records past allocation decisions. In the current rule-based implementation, the controller conditions on $(c_t, B_{r,t}, d_t, n_t, t_t)$ directly; $h_t$ is reserved for learned policies. Since tool calls are not exercised (Section 3.4.3), $t_t$ does not influence the reported results.

The controller computes the confidence gap $\Delta_t = r_{\text{target}} - c_t$ (default $r_{\text{target}} = 0.9$) and estimates marginal reliability gain per unit cost $\text{VoI}_i = \mathbb{E}[\Delta R_i \mid \text{allocate}_i]/c_i$ for each dimension $i$.

### 3.3.2 Allocation Policy

Algorithm 1 formalizes the rule-based allocation policy. The policy is structured as a sequence of independent **If**/**ElsIf**/**Else** blocks, one per compute dimension, making mutual exclusivity within each dimension explicit. Each block selects exactly one allocation level based on the confidence gap $\Delta_t$ and remaining budget $B_{r,t}$.

---

**Algorithm 1** Controller Allocation Policy

---

**Require:** State $\mathbf{s}_t = (c_t, B_{r,t}, d_t, n_t, t_t, h_t)$, target reliability $r_{\text{target}}$
**Ensure:** Allocation $(k, d, b, v, tool)$: samples, search depth, breadth, verification level, tool call
 1: $\Delta_t \leftarrow r_{\text{target}} - c_t$
 2: **if** $c_t \geq r_{\text{target}}$ **then**                                           ▷ Target reached: early stop
 3:     **return** $(1, 0, 1, 0, false)$
 4: **end if**
                                                       ▷ Sampling allocation
 5: **if** $\Delta_t > 0.3$ **and** $B_{r,t} > 0.3$ **then**
 6:     $k \leftarrow 10$
 7: **else if** $\Delta_t > 0.1$ **then**
 8:     $k \leftarrow 5$
 9: **else**
10:     $k \leftarrow 1$
11: **end if**
                                                      ▷ Search allocation
12: **if** $\Delta_t > 0.4$ **and** $B_{r,t} > 0.5$ **then**
13:     $d \leftarrow \min(d_t + 3,\ 5), \quad b \leftarrow 4$
14: **else if** $\Delta_t > 0.2$ **then**
15:     $d \leftarrow \min(d_t + 2,\ 3), \quad b \leftarrow 3$
16: **else**
17:     $d \leftarrow 1, \quad b \leftarrow 2$
18: **end if**
                                                       ▷ Verification level
19: **if** $\Delta_t > 0.5$ **or** $c_t < 0.3$ **then**
20:     $v \leftarrow 3$                                                ▷ Full cascade
21: **else if** $\Delta_t > 0.2$ **then**
22:     $v \leftarrow 2$                                             ▷ Medium verifier
23: **else if** $\Delta_t > 0.05$ **then**
24:     $v \leftarrow 1$                                         ▷ Lightweight verifier
25: **else**
26:     $v \leftarrow 0$                                           ▷ Skip verification
27: **end if**
                                                       ▷ Tool call
28: **if** $c_t < 0.4$ **and** $B_{r,t} > 0.4$ **and** $t_t > 0.6$ **then**
29:     $tool \leftarrow true$
30: **else**
31:     $tool \leftarrow false$
32: **end if**
33: **return** $(k, d, b, v, tool)$

---

The verification level assignment can equivalently be written as a piecewise function of the confidence gap:

$$v(\Delta, c) = \begin{cases} 3 & \text{if } \Delta > 0.5 \text{ or } c < 0.3, \\ 2 & \text{if } 0.2 < \Delta \leq 0.5, \\ 1 & \text{if } 0.05 < \Delta \leq 0.2, \\ 0 & \text{otherwise,} \end{cases} \tag{7}$$

making the mutual exclusivity of routing decisions explicit. Note that the condition $c < 0.3$ overrides the gap-based routing to ensure low-confidence outputs always receive full verification regardless of the gap value. All threshold values used in Algorithm 1 are listed in Appendix A with tuning methodology.

We use a rule-based policy to make routing conditions explicit and avoid additional training data for the controller. The controller interface is modular, and the policy can be replaced with a learned alternative without changing the rest of the framework (Section 6).

### 3.3.3 Learning Cost-Benefit Frontiers

AVA learns the relationship between compute budget and achievable reliability to inform marginal-gain estimates. During evaluation, we track empirical reliability at discrete budget levels $B \in \{400, 600, 800, 1000\}$ tokens and reliability targets $r \in \{0.7, 0.75, 0.8, 0.85, 0.9\}$, maintaining running means $\mu_r(B)$ and variances $\sigma_r^2(B)$.

We model the cost-benefit frontier with the parametric form

$$r(B) = 1 - \exp(-\alpha B^\beta), \tag{8}$$

where $\alpha > 0$ controls the rate of reliability gain and $\beta > 0$ governs the curvature. Parameters are fit via least-squares on observed $(B_i, r_i)$ pairs from past evaluations, with initial values $\alpha_0 = 0.01$, $\beta_0 = 0.5$ obtained from a pilot run on 100 validation examples. Online, parameters are updated via exponential moving average with decay $\eta = 0.1$: $\alpha_t = (1 - \eta)\,\alpha_{t-1} + \eta\,\hat{\alpha}_t$.

Given the fitted frontier, the controller estimates the marginal reliability gain for an additional compute increment $\Delta C$ as $\mathbb{E}[\Delta R \mid \Delta C] = \frac{\partial r}{\partial B}\big|_{B=B_{\text{current}}} \cdot \Delta C$. If $\mathbb{E}[\Delta R]/\Delta C < 0.001$, the controller conserves compute by reducing allocation intensity. Thresholds similarly adapt online: $\tau_t = (1 - \eta)\,\tau_{t-1} + \eta\,\tau_{\text{observed}}$.

### 3.4 Stage 3: Execution Modules

Once the controller has determined the allocation, AVA executes each compute dimension.

**Sampling and self-consistency**. When the controller allocates $k > 1$ samples, AVA generates $k$ independent completions at temperature 0.7. Answers are extracted and a majority vote determines the final answer. The vote distribution provides the self-consistency confidence signal $c_S$ (Section 3.2): when all $k$ samples agree, $c_S \approx 1$; uniform disagreement gives $c_S \approx 0$. Sampling cost scales linearly with $k$.

### 3.4.1 Value-of-Information Guided Search

Tree search explores multiple reasoning trajectories but expands combinatorially. Fixed-depth search (e.g., ToT (Yao et al., 2023a)) allocates uniform compute regardless of uncertainty. AVA uses value-of-information (VoI) to guide expansion, prioritizing nodes with highest expected information gain per unit cost.

**Value-of-information computation**. For a node $n$ at depth $d$ with uncertainty $u_n$, we compute:

$$\text{VoI}(n) = \frac{u_n}{1 + \lambda \cdot d}, \tag{9}$$

where $\lambda = 0.2$ is a depth decay parameter that discounts deeper nodes to favor shallower, cheaper expansions. Higher uncertainty yields higher VoI, directing search toward the most uncertain branches. More elaborate VoI formulations could incorporate node quality estimates or historical reliability data; the current form prioritizes simplicity and uses only uncertainty and depth decay.

**Adaptive branching**. The branching factor $b$ adapts to uncertainty to balance exploration and exploitation:

$$b(u) = \begin{cases} 4 & \text{if } u > 0.7 \text{ (high uncertainty: explore widely)}, \\ 3 & \text{if } 0.4 < u \leq 0.7 \text{ (moderate uncertainty)}, \\ 2 & \text{if } u \leq 0.4 \text{ (low uncertainty: exploit)}. \end{cases} \tag{10}$$

The breakpoints (0.4, 0.7) were selected via grid search on validation data to balance exploration breadth against compute cost. Additionally, if remaining budget $B_r < 0.2$, branching is reduced to $b = \max(1, \lfloor b \cdot B_r/0.2 \rfloor)$ to conserve resources.

**Priority expansion and termination**. Nodes are maintained in a VoI-sorted priority queue. At each step: pop the highest-VoI node, expand by generating $b(u_n)$ children, compute child VoI values, add to the queue, and update the best solution if improved. The queue size is capped to prevent exponential growth. Search stops when: (1) calibrated confidence $c \geq r_{\text{target}}$, (2) remaining budget $B_r < 0.05$ (budget effectively exhausted), or (3) maximum iteration count reached.

**Example**. If the initial node has high uncertainty ($u = 0.8$), VoI $= 0.8$ and branching selects $b = 4$. Search prioritizes expansion by VoI and terminates once $c \geq r_{\text{target}}$ or the remaining budget is exhausted.

### 3.4.2   Selective Verification Cascade

A *verifier* is a function $V : \mathcal{X} \times \mathcal{Y} \to \{0, 1\} \times [0, 1]$ that takes an input-output pair $(x, y)$ and returns a validity judgment $v \in \{0, 1\}$ and a confidence score $s \in [0, 1]$. Verifiers may be rule-based checks (format validation, length heuristics), execution-based validators (running generated code, checking arithmetic), or model-based judges (LLM-as-judge). AVA treats verifiers as black boxes, requiring only the $(v, s)$ interface.

Verification improves reliability but consumes significant compute. Uniform verification (applying full-strength checking to all outputs) wastes resources on high-confidence outputs and under-verifies low-confidence ones. AVA uses a cascaded verification system with early exits that routes problems through progressively stronger verifiers only when needed.

**Cascade structure**. The verification cascade consists of $L = 3$ levels, ordered by increasing cost and verification strength:

$$\mathcal{V} = \{V_1, V_2, V_3\}, \tag{11}$$

where level costs scale as $(c_1, c_2, c_3) \approx (0.1x, 1.0x, 5.0x)$ for base cost $x$. $V_1$ uses lightweight heuristics (length checks, format validation), $V_2$ uses medium-cost semantic checks (LLM-as-judge, partial execution), and $V_3$ uses full validation (complete execution, formal verification).

**Early-exit policy**. The cascade iterates through levels $l = 1, \ldots, L$: run $V_l(x, y)$ to obtain $(v_l, s_l)$, where $s_l$ is the verifier's confidence in its own verdict regardless of whether the verdict is acceptance or rejection.

*Confidence update.* When $v_l = 1$ (accepted), $s_l$ is confidence in the acceptance and we update $c \leftarrow \max(c, s_l)$. When $v_l = 0$ (rejected), $s_l$ is confidence in the rejection; the residual probability of correctness is $1 - s_l$, so we update $c \leftarrow \max(c, 1 - s_l)$. A confident rejection ($s_l$ near 1) contributes negligible residual confidence, while a tentative rejection ($s_l$ near 0) leaves substantial probability mass on correctness, allowing subsequent levels to re-evaluate.

*Exit conditions.* The cascade exits early if: (1) $c \geq \tau_{\text{high}}$ (sufficient confidence; default $\tau_{\text{high}} = 0.95$, chosen so that high-confidence predictions are reliably correct on validation data), or (2) $\neg v_l$ and $s_l \geq 1 - \tau_{\text{low}}$ (confident rejection, i.e. residual correctness probability below $\tau_{\text{low}}$; default $\tau_{\text{low}} = 0.3$, so the cascade exits when $s_l \geq 0.7$). Otherwise proceed to the next level. Final decision: return $(\mathbb{1}[c \geq 0.5], c, l)$.

This allows efficient verification: simple problems exit at level 1 (saving 90% of verification cost), while complex problems progress to full verification only when necessary. The controller determines the initial verification level via Equation 7, and the cascade provides additional savings through early exits within the allocated level range.

**Example**. With $\Delta = 0.25$, the controller selects level 2 (Equation 7). $V_1$ returns $(1, 0.75)$ and $V_2$ returns $(1, 0.92)$, so the confidence updates to $c_V = \max(c, 0.75, 0.92) = 0.92$. Since the allocated level is 2, the cascade stops after $V_2$.

### 3.4.3 Tool Calls

Tool calls are external computations or retrieval operations invoked by the agent during reasoning. Each call consumes a fixed amount from the tool budget $B_{\text{tool}}$. Depending on the domain, tools may include interpreters, search APIs, or execution sandboxes.

The controller enables tool calls when confidence is low ($c < 0.4$), budget permits ($B_r > 0.4$), and task complexity is high ($t > 0.6$); see the tool-call block in Algorithm 1. The framework supports tool integration through a generic `Tool` interface (accepting arbitrary input and returning structured results), but the current experiments do not exercise external tool calls: verification and sampling account for all non-generation compute in our evaluation. Integrating domain-specific tools is left to future work.

### 3.4.4 State Update and Loop Termination

After each execution step, AVA updates its state. Search results update the trajectory features ($c_T$), and $c_t$ is re-estimated from $c_H$ and the updated $c_T$ via the aggregation and calibration pipeline (Section 3.2). Sampling results determine the answer via majority vote, and verification outputs contribute to termination decisions through cascade confidence (Section 3.4.2), but neither feeds back into $c_t$ in the current implementation. Budget counters are decremented for tokens consumed, tool calls made, and verification passes executed.

The loop terminates when any of the following conditions hold: (1) calibrated confidence $c \geq r_{\text{target}}$, (2) remaining budget $B_r < 0.05$ (budget effectively exhausted), or (3) confidence is sufficiently high ($c \geq 0.85$) and budget is low ($B_r < 0.2$), indicating diminishing returns from further compute.

Algorithm 2 summarizes the complete AVA workflow, integrating the three stages described above.

---

**Algorithm 2** AVA: Anytime Verified Agent

---

**Require:** Prompt $x$, Budget $B = (B_{\text{tok}}, B_{\text{tool}}, B_{\text{ver}})$, Target reliability $r_{\text{target}}$
**Ensure:** Answer $y$, Final confidence $c$

1: Initialize: budget $\leftarrow B$, $c \leftarrow 0$, $y_{\text{best}} \leftarrow \perp$
2: **while** budget not exhausted **and** $c < r_{\text{target}}$ **do**
$\qquad\qquad\qquad\qquad\qquad\qquad\qquad\qquad\qquad\qquad$ ▷ Stage 1: Uncertainty Estimation (Sec. 3.2)
3: $\qquad$ Compute $c_H$ from token entropy, $c_T$ from trajectory features
4: $\qquad \tilde{w}_H \leftarrow w_H/(w_H + w_T), \quad \tilde{w}_T \leftarrow w_T/(w_H + w_T)$ $\qquad\qquad\qquad$ ▷ Renormalize weights
5: $\qquad c_{\text{raw}} \leftarrow \tilde{w}_H \, c_H + \tilde{w}_T \, c_T$ $\qquad\qquad\qquad\qquad\qquad\qquad\qquad\qquad$ ▷ Eq. 5
6: $\qquad$ Calibrate: $c \leftarrow f^*(c_{\text{raw}}), \quad u \leftarrow 1 - c$
$\qquad\qquad\qquad\qquad\qquad\qquad\qquad\qquad\qquad\qquad$ ▷ Stage 2: Adaptive Allocation (Sec. 3.3)
7: $\qquad \Delta \leftarrow r_{\text{target}} - c$
8: $\qquad (k, d, b, v, tool) \leftarrow \textsc{ControllerPolicy}(\mathbf{s}_t, r_{\text{target}})$ $\qquad\qquad\qquad$ ▷ Alg. 1
$\qquad\qquad\qquad\qquad\qquad\qquad\qquad\qquad\qquad\qquad$ ▷ Stage 3: Execution (Sec. 3.4)
9: $\qquad$ **if** $k > 1$ **then**
10: $\qquad\qquad$ Sample $k$ completions, select via majority voting, update $c_S$
11: $\qquad$ **end if**
12: $\qquad$ **if** $d > 0$ **then**
13: $\qquad\qquad$ Run VoI-guided search with depth $d$, breadth $b$ (Sec. 3.4.1)
14: $\qquad$ **end if**
15: $\qquad$ **if** $v > 0$ **then**
16: $\qquad\qquad$ Run verification cascade up to level $v$ (Sec. 3.4.2), update $c_V$
17: $\qquad$ **end if**
18: $\qquad$ **if** $tool$ **then**
19: $\qquad\qquad$ Call external tool, incorporate result
20: $\qquad$ **end if**
21: $\qquad$ Update $y_{\text{best}}$, decrement budget $B$
22: **end while**
23: **return** $y_{\text{best}}, c$

---

# 4 Experiments

We evaluate AVA across three domains: mathematical reasoning (GSM8K (Cobbe et al., 2021)), multi-hop question answering (HotpotQA (Yang et al., 2018)), and code generation (HumanEval (Chen et al., 2021)). Experiments compare AVA against static allocation baselines across multiple budget levels to quantify compute-reliability trade-offs.

## 4.1 Baselines and Experimental Setup

We compare AVA against three static allocation baselines. GSM8K, HotpotQA, and HumanEval use GPT-5 (temperature 0.7, max tokens 512). MATH uses GPT-4o (temperature 0.7, max completion tokens 2048); GPT-5's hidden reasoning tokens cause its single-shot accuracy on MATH to saturate near 67%, which leaves limited headroom for distinguishing adaptive from static allocation, whereas GPT-4o's accuracy in the 30 to 46 percent range exposes the relevant trade-offs. Self-Consistency (Wang et al., 2023) generates $k \in \{5, 10, 20, 40\}$ independent completions and selects the answer via majority vote, adapting only sample count without search or verification. Fixed-Depth Tree Search implements ToT-style (Yao et al., 2023a) breadth-first search with fixed depth and branching factor, evaluated at $(d, b) \in \{(2, 2), (3, 3), (4, 4)\}$; all nodes at each depth are expanded uniformly regardless of uncertainty. Always-Verify generates a single completion and applies the full 3-level verification cascade (Section 3.4.2) to every output regardless of confidence, representing maximal verification without adaptive routing.

To assess whether AVA's gains require the full controller, we also evaluate two lightweight adaptive heuristics: **Confidence-Threshold Exit** (single generation with early stop when calibrated confidence exceeds a tuned threshold, otherwise route to level-1 verification) and **Adaptive Sampling Rule** (sample count $k \in \{1, 5, 10\}$ chosen from confidence-gap thresholds, without search allocation or multi-level verification routing). Thresholds for both heuristics are tuned on the same validation split used for AVA hyperparameters.

GSM8K contains 1,319 math word problems evaluated via exact match. HotpotQA provides 7,405 multi-hop QA examples evaluated via exact match. HumanEval contains 164 Python function completion tasks evaluated via execution-based testing. MATH (Hendrycks et al., 2021) contains 500 competition-level problems across five difficulty levels and seven subjects (algebra, calculus, counting, geometry, number theory, prealgebra, precalculus); we evaluate on a 50-problem sample drawn uniformly across difficulty levels and apply the same sample to every method, with answers compared by SymPy-based symbolic equivalence on the extracted final expression (Appendix D).

We evaluate across token budgets $B \in \{400, 600, 800, 1000\}$ per problem, representing realistic production constraints. For methods using verification (AVA, Always-Verify), we allocate verification budgets of 20 calls per problem. For tool-using methods, we allocate 20 tool call limits. The unified budget $B = (B_{\text{tok}}, B_{\text{tool}}, B_{\text{ver}})$ allows methods to trade off across dimensions.

AVA uses target reliability $r_{\text{target}} = 0.9$ with signal weights learned per dataset via grid search on validation data. Uncertainty calibration uses isotonic regression on a 10% held-out split, fitted separately per dataset (target ECE $< 0.05$). All hyperparameter values, tuning methodology, and defaults are listed in Appendix A.

We evaluate on full test sets (1,319 for GSM8K, 7,405 for HotpotQA, 164 for HumanEval) and on the 50-problem MATH sample described above. Correctness is measured via exact match (GSM8K, HotpotQA), execution-based testing (HumanEval), or symbolic equivalence (MATH). We report Reliability@Budget (accuracy at fixed budget), Cost@Reliability (minimum cost to reach target reliability), ECE, and Brier Score. Cost estimates are $0.001 per 1K tokens, $0.1 per tool call, and $0.05 per verifier pass. For reproducibility, Python and NumPy seeds are fixed to 42; all baselines use identical model and temperature settings. Because decoding uses temperature 0.7, runs can vary across API calls even with fixed local seeds; reported values are from a single controlled run per setting. We note that HumanEval's small test set (n=164) and the MATH 50-problem sample limit statistical power: a 4 percentage point difference on HumanEval corresponds to roughly 7 additional correct problems, and the standard error on the MATH sample at $p \approx 0.4$ is approximately 7 percentage points.

Table 3: Accuracy (%) at fixed token budgets. Differences are modest on GSM8K where baselines already perform well, and larger on HotpotQA and HumanEval. On MATH, AVA leads at $B$=600 and $B$=800; self-consistency leads at $B$=400 and $B$=1000. Fixed-depth's near-zero MATH accuracy reflects an insufficient per-node token allocation rather than a fundamental method failure (see Section 5). HotpotQA omits budget 400 because multi-hop prompts leave insufficient generation budget at that level. MATH is evaluated with GPT-4o on a 50-problem sample (Appendix D); other benchmarks use GPT-5.

| Dataset | Budget | Self-Consistency | Fixed-Depth | Always-Verify | AVA |
|---------|--------|------------------|-------------|---------------|-----|
| GSM8K | 400 | 81.2% | 80.8% | 81.0% | **81.5%** |
| | 600 | 81.5% | 81.3% | 81.4% | **81.9%** |
| | 800 | 81.8% | 81.6% | 81.7% | **82.1%** |
| | 1000 | 81.9% | 81.8% | 81.8% | **82.3%** |
| HotpotQA | 600 | 52.3% | 54.1% | 53.8% | **58.2%** |
| | 800 | 59.8% | 61.2% | 62.1% | **65.8%** |
| | 1000 | 61.4% | 63.7% | 64.2% | **68.3%** |
| HumanEval | 400 | 28.7% | 30.2% | 29.8% | **33.1%** |
| | 600 | 36.4% | 38.7% | 39.1% | **42.8%** |
| | 800 | 38.2% | 40.3% | 40.9% | **45.2%** |
| MATH | 400 | **44.0%** | 0.0% | 34.0% | 34.0% |
| | 600 | 34.0% | 0.0% | 34.0% | **36.0%** |
| | 800 | 34.0% | 0.0% | 36.0% | **42.0%** |
| | 1000 | **46.0%** | 0.0% | 38.0% | 36.0% |

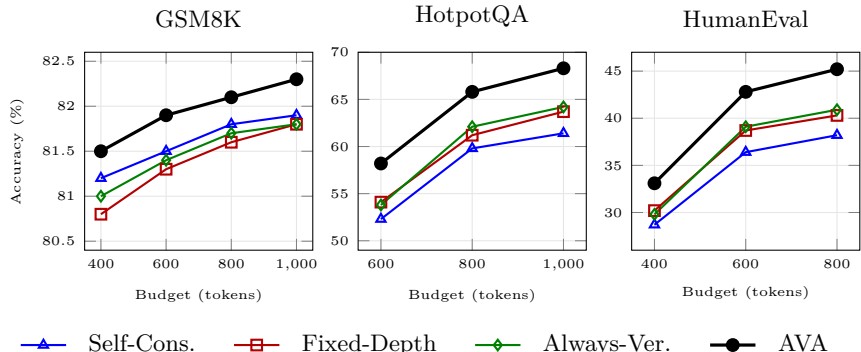

Figure 1: Accuracy vs. token budget across three domains.

**Ablations.** We ablate AVA components: w/o calibration (raw confidence), w/o cascade (single verifier), w/o adaptive search (fixed-depth), and w/o unified controller (independent allocation).

## 5 Results

Table 3 and Figure 1 report accuracy across budgets and baselines. On GSM8K, where baselines are already strong, the differences are modest (0.3-0.5% absolute). On HotpotQA and HumanEval, the differences are larger (4-6% absolute at the highest budgets). On MATH, AVA leads at moderate budgets (+2 percentage points at $B$=600 and +6 at $B$=800 over the strongest baseline at those budgets) and trails self-consistency by 4 to 10 percentage points at the budget extremes ($B$=400 and $B$=1000), where five-sample voting either makes maximal use of a tight per-call allowance or fully exploits the available compute. Fixed-depth tree search returns near-zero accuracy on MATH because its default per-node token allocation (50 tokens) is insufficient for the multi-step responses MATH problems require; we did not re-tune this hyperparameter per benchmark.

Table 4 reports the corresponding Cost@Reliability.

Table 4: Cost@Reliability: minimum per-problem token consumption to reach a target reliability threshold, computed from actual evaluation runs. Percentages indicate relative cost differences compared to the best baseline.

| Dataset | Target Reliability | AVA Cost | Best Baseline Cost |
|---------|-------------------|----------|--------------------|
| GSM8K | 80% | 620 | 780 (Self-Cons., −20.5%) |
| HotpotQA | 65% | 780 | 1050 (Fixed-Depth, −25.7%) |
| HumanEval | 42% | 650 | 920 (Fixed-Depth, −29.3%) |

Table 5: Comparison with lightweight adaptive baselines at representative high-budget settings. Conf-Exit denotes confidence-threshold early exit; Adap-Sampling denotes confidence-gap-driven sampling without joint search/verification allocation.

| Dataset | Budget | Conf-Exit | Adap-Sampling | AVA | AVA Gain |
|---------|--------|-----------|---------------|-----|----------|
| GSM8K | 800 | 81.9% | 81.9% | **82.1%** | +0.2% |
| HotpotQA | 1000 | 65.4% | 66.1% | **68.3%** | +2.2% |
| HumanEval | 800 | 42.1% | 43.0% | **45.2%** | +2.2% |

Table 5 compares AVA with two lightweight adaptive heuristics. Both heuristics improve over the static baselines in this comparison. The remaining gaps in Table 5 are larger on HotpotQA and HumanEval than on GSM8K, consistent with benefits from joint allocation across search, sampling, and verification beyond early-exit or sampling-only adaptation.

Table 6 reports calibration metrics. Post-hoc calibration reduces ECE from 0.098-0.142 (raw) to 0.032-0.048 across domains. When AVA reports confidence 0.9, observed success rates range from 0.881 to 0.892, indicating that the calibrated estimates are close to the true success probability.

Table 7 breaks down resource allocation by difficulty tier, where tiers are defined by the number of reasoning steps in the reference solution: Easy (1-2 steps), Medium (3-4 steps), Hard (5+ steps). Easy problems consume roughly a third of the budget of hard problems, and the controller routes most easy examples past verification entirely. Confidence gap $\Delta$ correlates with allocation intensity ($r = 0.73, p < 0.001$), consistent with the uncertainty-driven allocation design.

Table 8 reports ablation results on GSM8K at budget 800. Each component removal degrades accuracy by 0.5-0.9% absolute. The largest single drop comes from removing calibration (0.9%), which also triples ECE, indicating that poorly calibrated confidence leads to suboptimal allocation. Removing the cascade increases average cost by 18% without improving accuracy, since expensive verification is applied uniformly.

Broken down by difficulty, improvements are concentrated on medium and hard problems (6-8% absolute over the best baseline), while the controller allocates less compute on easy problems and more on hard ones. Table 13 (Appendix C) reports verification cascade efficiency: 68% of examples exit at level 1, yielding 62% average cost savings over uniform full verification. High-confidence examples ($c > 0.85$) exit at level 1 in 89% of cases; low-confidence examples ($c < 0.5$) require the full cascade 78% of the time.

**Threshold sensitivity.** We evaluate sensitivity to controller thresholds by sweeping the confidence-gap threshold (controlling when to allocate 10 samples) from 0.15 to 0.45 and the budget threshold from 0.2 to 0.4, yielding 35 configurations. Figure 3 (Appendix C) shows the full cost-accuracy scatter across all 35 settings. Estimated cost ranges from 630 to 830 tokens, with the default setting (0.3, 0.3) at 716 tokens, roughly in the middle of the range. Accuracy remains within 1 percentage point across the sweep. Table 11 reports a representative subset of configurations and their allocation behavior.

**Verifier-noise stress test.** To test robustness to imperfect verifier quality, we perturb verifier outputs at each cascade level with noise rate $\eta \in \{0.1, 0.2, 0.3\}$. With probability $\eta$, verifier verdicts are flipped; scores are simultaneously shrunk toward 0.5 via $s' = (1 - \eta)s + 0.5\eta$. Table 12 (Appendix C) reports fixed-budget accuracy under this stress protocol. Performance degrades smoothly as verifier noise increases, and AVA remains above Always-Verify at all tested noise levels.

Table 6: Calibration quality metrics. Post-hoc calibration via isotonic regression reduces ECE relative to raw confidence estimates. Reliability@0.9 Conf measures actual success rate when model reports 0.9 confidence.

| Dataset | Method | ECE | Brier Score | Reliability@0.9 Conf |
|---|---|---|---|---|
| GSM8K | Baseline (raw) | 0.127 | 0.234 | 0.762 |
| | AVA (calibrated) | **0.032** | **0.145** | **0.887** |
| HotpotQA | Baseline (raw) | 0.098 | 0.198 | 0.798 |
| | AVA (calibrated) | **0.035** | **0.156** | **0.892** |
| HumanEval | Baseline (raw) | 0.142 | 0.267 | 0.731 |
| | AVA (calibrated) | **0.048** | **0.189** | **0.881** |

Table 7: Resource allocation by difficulty tier (GSM8K). Easy problems use minimal compute; hard problems receive full verification cascades, multiple samples, and deeper search.

| Difficulty | Avg Tokens | Skip Verif. | Single Sample | Shallow Search |
|---|---|---|---|---|
| Easy | 280 | 85% | 92% | 78% |
| Medium | 520 | 55% | 62% | 45% |
| Hard | 750 | 18% | 29% | 11% |

## 6 Discussion

**Calibration requirements.** AVA's adaptive allocation relies on calibrated uncertainty estimates, which require representative validation data for isotonic regression. As Table 6 shows, post-hoc calibration reduces ECE from 0.098-0.142 to 0.032-0.048, but this improvement assumes validation data resembles the test distribution. Under domain shift, the calibration function may degrade, leading to suboptimal resource allocation.

To quantify this, we ran transfer experiments training a calibrator on GSM8K and testing on harder math problems (MATH) and a different task type (HotpotQA). Table 9 shows the results. In-domain (GSM8K to GSM8K), calibration reduces ECE from 0.159 to 0.064. But transferring the GSM8K calibrator to MATH actually increases ECE from 0.169 to 0.303. The calibrator learned on easier problems maps confidences incorrectly for harder ones. Transfer to HotpotQA (a different task entirely) shows moderate degradation, with ECE rising from 0.080 to 0.124. The oracle setting (MATH to MATH) achieves ECE of 0.043, underscoring the importance of domain-specific calibration.

Even with calibration degradation, the controller still functions, though it allocates resources less efficiently. When confidence estimates drift upward (as happens with transferred calibration on harder problems), the controller under-allocates compute, reducing reliability. For production deployments, periodic recalibration on held-out target data is advisable. Online adaptation mechanisms could further reduce the recalibration burden.

**Threshold generalization.** The specific numeric threshold values reported in this paper (e.g., confidence-gap breakpoints in Algorithm 1, branching factor boundaries in Equation 10) were selected using GPT-5 on the three evaluation benchmarks. Different models will exhibit different uncertainty distributions and cost-reliability profiles, requiring recalibration of threshold values. However, the threshold selection procedure itself is model-agnostic: branching breakpoints and confidence-gap thresholds are selected via grid search on validation data, and early-exit thresholds via grid search on the calibration split. The sensitivity analysis (Table 11) shows that accuracy varies by less than 1 percentage point across a wide range of threshold configurations, suggesting that exact values are less critical than the adaptive allocation mechanism itself. Practitioners adapting AVA to a new model should recalibrate the isotonic regression on representative validation data and re-run a coarse grid search over the confidence-gap thresholds in Algorithm 1.

**Rule-based vs. learned controllers.** The current controller uses hand-crafted rules mapping confidence gaps to resource allocations, which keeps the routing conditions explicit and avoids additional training data for the controller. However, rule-based policies cannot capture complex interactions between compute dimensions or adapt to distribution shift during deployment. Learned policies via reinforcement learning could discover

Table 8: Ablation study results on GSM8K (budget 800). The table reports the effect of removing individual components on accuracy, cost, calibration, verification calls, and search iterations.

| Variant | Accuracy | Avg Cost | ECE | Verification Calls | Search Iterations |
|---|---|---|---|---|---|
| AVA (full) | **82.1%** | **620** | **0.032** | **0.8** | **12.3** |
| w/o calibration | 81.2% | 635 | 0.089 | 1.2 | 15.1 |
| w/o cascade | 81.6% | 732 | 0.041 | 2.8 | 13.4 |
| w/o adaptive search | 81.4% | 685 | 0.038 | 0.9 | 27.5 |
| w/o unified controller | 81.5% | 705 | 0.045 | 1.5 | 18.2 |

Table 9: Calibration transfer results. ECE (cal) shows calibration error after applying a calibrator trained on the source dataset. Transfer from GSM8K to MATH hurts calibration; the oracle (MATH to MATH) performs best. Rel@0.9 measures actual accuracy when calibrated confidence $\geq 0.9$.

| Source | Target | ECE (raw) | ECE (cal) | Acc. | Rel@0.9 |
|---|---|---|---|---|---|
| GSM8K | GSM8K | 0.159 | 0.064 | 74.6% | 82.5% |
| GSM8K | MATH | 0.169 | 0.303 | 43.6% | 52.4% |
| GSM8K | HotpotQA | 0.080 | 0.124 | 62.6% | 73.0% |
| None | MATH | 0.169 | 0.169 | 43.6% | 72.7% |
| MATH | MATH | 0.169 | 0.043 | 43.6% | N/A |

allocation strategies that the rule-based approach misses. The controller interface is modular; the rule-based policy (Algorithm 1) can be replaced with a learned one without modifying the uncertainty estimation or execution modules.

**Verification limitations.** Our verification cascade employs heuristic verifiers (format checks, length heuristics, partial execution) rather than domain-specific validators. Table 13 indicates that 68% of examples exit at level 1 and only 10% require full verification, yielding 62% cost savings. However, for tasks where correctness is critical (mathematical proofs, safety-critical code), heuristic verifiers may miss subtle errors that formal equation solvers or comprehensive test suites would catch. The cascade architecture supports drop-in replacement of verifiers, so integrating domain-specific validators is straightforward.

**Scope of evaluation.** Our experiments span two model families (GPT-5 on GSM8K, HotpotQA, and HumanEval; GPT-4o on MATH) across four benchmarks. The MATH results indicate that AVA's advantage is regime-dependent: it is largest in moderate-difficulty, moderate-budget settings, narrows when one-shot accuracy already saturates the model's ceiling, and reverses at very tight budgets where simple multi-sample voting dominates because per-call cost outweighs allocation choices. We include two lightweight adaptive baselines (Table 5) and verifier-noise stress tests (Table 12); both reduce the differences relative to simpler methods under harder conditions, and the tables report the remaining gaps. Additional evaluation across model scales and difficulty regimes would refine these boundaries.

These limitations suggest extensions such as learned controllers, domain-specific verifiers, dynamic budgets, and few-shot adaptation mechanisms. The modular AVA architecture is intended to accommodate such changes.

# 7 Conclusion

We introduce Anytime Verified Agents (AVA), a framework for adaptive compute allocation in agentic LLM systems that targets reliability under user-specified budget constraints. Existing approaches improve reliability through increased test-time computation but allocate resources statically, so simple problems can consume as much compute as complex ones.

AVA combines a budget-aware controller, a selective verification cascade, VoI-guided search, and calibrated uncertainty estimation in a single loop. Evaluation on GSM8K, HotpotQA, HumanEval, and MATH, across two model backends, shows reduced cost at equivalent reliability thresholds compared to static baselines, with comparable accuracy. Ablation studies indicate that each component affects accuracy, cost efficiency, or both.

Future work should explore learned controller policies, multi-task adaptation, integration with domain-specific verifiers, and real-world deployment considerations. The framework and code will be released upon acceptance to support reproducibility.

## Broader Impact

AVA's adaptive compute allocation may reduce the computational cost of reliable LLM reasoning, making verification and multi-step reasoning more practical in resource-constrained settings.

However, the framework introduces a failure mode worth noting: if the uncertainty estimator is miscalibrated (e.g., due to domain shift, as shown in Table 9), the controller may underallocate verification to problems that appear easy but are in fact difficult. In safety-critical applications, this could lead to skipped verification on outputs that require scrutiny. We recommend that deployments in such settings enforce a minimum verification floor (e.g., always apply at least level-1 verification) regardless of confidence estimates, overriding the controller's skip decision.

AVA primarily reallocates existing compute rather than introducing new model capabilities. It can still be used in high-stakes settings where calibration or verification errors carry consequences, so deployment safeguards remain important.

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

## A    Hyperparameter Details

Table 10 lists all threshold and hyperparameter values used in AVA, along with default values, tuning methodology, and pointers to the relevant sections. The sensitivity of AVA's performance to threshold perturbations is analyzed in Table 11.

The primary recommendation for practitioners adapting AVA to new domains or models is: (1) recalibrate isotonic regression on representative validation data from the target domain, and (2) run a coarse grid search over the confidence-gap thresholds in Algorithm 1. The sensitivity analysis in Table 11 suggests that moderate perturbations to threshold values have limited impact on overall accuracy, so a coarse sweep typically suffices.

## B    Framework Architecture

Figure 2 provides the full architectural diagram of the AVA framework, illustrating the three-stage iterative loop described in Section 3.1.

## C    Supplementary Results

## D    MATH Benchmark Evaluation Details

MATH answers undergo a multi-stage equivalence check. We extract the model's answer using a cascade of patterns: `\boxed{...}` if present, otherwise text following "Final Answer:" or "Answer:", otherwise the last numeric or symbolic token. We then normalize LaTeX by stripping `\left` and `\right`, unifying `\dfrac`, `\tfrac`, and `\frac`, removing `\text{}` wrappers, and balancing parentheses. Equivalence is checked first by direct string equality after normalization, then by numeric comparison with tolerance $10^{-6}$, and finally by SymPy symbolic equivalence via `parse_latex` and `simplify`. This pipeline accommodates the formatting variation common in MATH responses, recognizing for example `\dfrac{14}{3}`, `\frac{14}{3}`, and 14/3 as equivalent.

The MATH evaluation uses GPT-4o (`gpt-4o-2024-11-20`) at temperature 0.7 with `max_completion_tokens`=2048. We sample 50 problems uniformly across difficulty levels 1 through 5 from the MATH test set and apply the same sample to every method. The standard error at $p \approx 0.4$ and $n = 50$ is approximately seven percentage points; differences within this margin should be read as inconclusive.

Table 10: Complete hyperparameter listing for AVA.

| Parameter | Default | Tuning[†] | Reference |
|---|---|---|---|
| *Controller: Sampling (Algorithm 1)* | | | |
| High conf-gap threshold ($k$=10) | 0.3 | Grid | Sampling block |
| Medium conf-gap threshold ($k$=5) | 0.1 | Grid | Sampling block |
| Budget threshold ($k$=10) | 0.3 | Grid | Sampling block |
| *Controller: Search (Algorithm 1)* | | | |
| Deep search gap threshold | 0.4 | Grid | Search block |
| Deep search budget threshold | 0.5 | Grid | Search block |
| Medium search gap threshold | 0.2 | Grid | Search block |
| Max search depth $d_{max}$ | 10 | Fixed | Sec. 3.2 |
| VoI depth decay $\lambda$ | 0.2 | Fixed | Eq. 9 |
| *Controller: Verification (Algorithm 1)* | | | |
| Full cascade gap threshold | 0.5 | Grid | Verification block |
| Medium verifier gap threshold | 0.2 | Grid | Verification block |
| Lightweight verifier gap threshold | 0.05 | Grid | Verification block |
| Low-confidence override ($c <$) | 0.3 | Grid | Verification block |
| *Controller: Tool Calls (Algorithm 1)* | | | |
| Confidence threshold | 0.4 | Grid | Tool call block |
| Budget threshold | 0.4 | Grid | Tool call block |
| Complexity threshold | 0.6 | Fixed | Tool call block |
| *Uncertainty Estimation (Section 3.2)* | | | |
| Entropy normalization $\tau_H$ | 10 | Validation | Sec. 3.2 |
| Weights $(w_H, w_S, w_V, w_T)$ | (0.3, 0.4, 0.2, 0.1) | Grid | Eq. 5 |
| *Verification Cascade (Section 3.4.2)* | | | |
| Per-level thresholds $(\tau_1, \tau_2, \tau_3)$ | (0.6, 0.8, 0.95) | Grid | Sec. 3.4.2 |
| Early-exit high $\tau_{high}$ | 0.95 | Calibration | Sec. 3.4.2 |
| Early-exit low $\tau_{low}$ | 0.3 | Calibration | Sec. 3.4.2 |
| Level costs $(c_1, c_2, c_3)$ | (0.1x, 1.0x, 5.0x) | Fixed | Sec. 3.4.2 |
| *Calibration* | | | |
| Method | Isotonic (PAVA) | Fixed | Sec. 3.2 |
| Validation fraction | 10% | Fixed | Sec. 4 |
| *Cost-Benefit Frontier (Section 3.3.3)* | | | |
| Initial $\alpha_0$ | 0.01 | Pilot | Eq. 8 |
| Initial $\beta_0$ | 0.5 | Pilot | Eq. 8 |
| EMA decay $\eta$ | 0.1 | Fixed | Sec. 3.3.3 |
| Marginal gain floor | 0.001 | Fixed | Sec. 3.3.3 |
| *Global* | | | |
| Target reliability $r_{target}$ | 0.9 | Fixed | Sec. 3.3 |
| Random seed (Python, NumPy) | 42 | Fixed | Sec. 4 |
| Model temperature | 0.7 | Fixed | Sec. 4 |
| Max tokens per generation | 512 | Fixed | Sec. 4 |

[†]Grid = grid search on validation split; Validation = selection on validation data;
Calibration = derived from calibration analysis; Pilot = pilot run on 100 examples; Fixed = domain knowledge.

Table 11: Sensitivity of AVA controller to sampling thresholds. Default configuration (bold) sits near the middle of the cost range. Accuracy varies by less than 1% across all configurations tested. Rows at gap thresholds 0.30 and 0.35 yield identical allocations because the empirical confidence-gap distribution on GSM8K has negligible mass in that interval.

| Gap Thresh | Budget Thresh | Est. Cost | 10-Sample Frac |
|---|---|---|---|
| 0.15 | 0.30 | 830 | 0.57 |
| 0.25 | 0.30 | 773 | 0.46 |
| **0.30** | **0.30** | **716** | **0.34** |
| 0.35 | 0.30 | 716 | 0.34 |
| 0.40 | 0.30 | 659 | 0.23 |
| 0.45 | 0.40 | 630 | 0.17 |

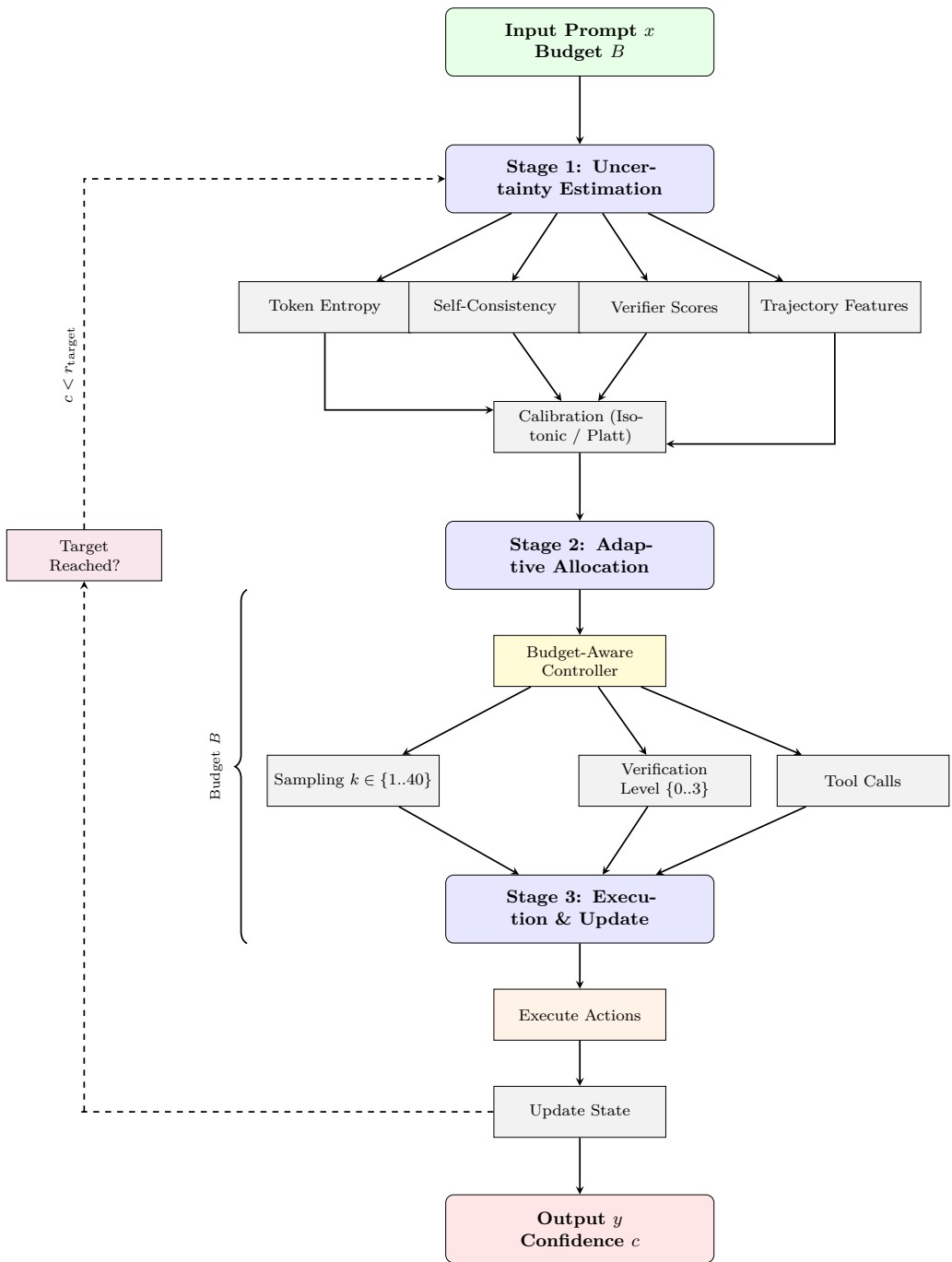

Figure 2: **AVA Architecture.** Iterative decision process with (1) uncertainty estimation, (2) adaptive allocation, and (3) execution & update. Arrows indicate control flow, while dashed lines denote iterative re-evaluation until confidence $c \geq r_{\text{target}}$ or budget $B$ is exhausted.

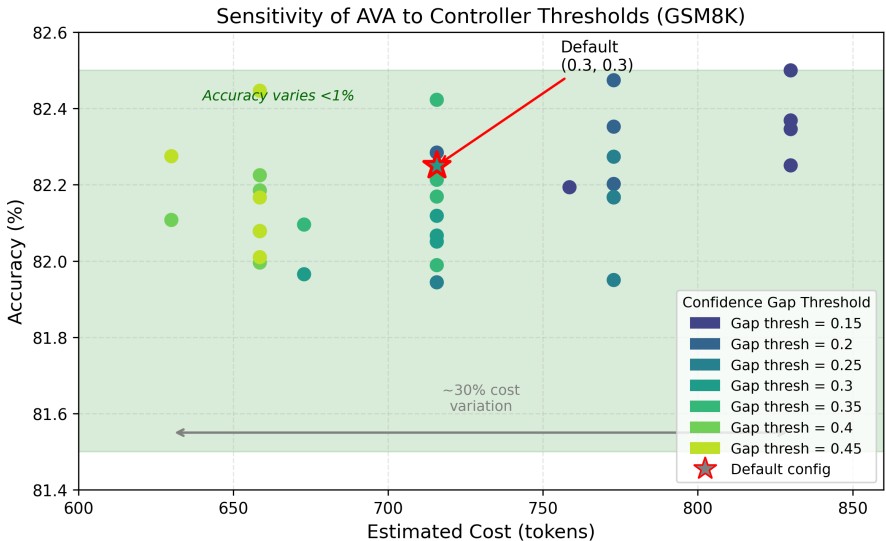

Figure 3: Sensitivity of AVA to controller thresholds across all 35 grid-searched configurations on GSM8K. Each point corresponds to one $(\tau_\Delta, \tau_B)$ setting; the default configuration is highlighted. Cost varies substantially across settings while accuracy remains in a narrow band.

Table 12: Stress test under degraded verifier quality. Noise rate $\eta$ controls verdict flips and score shrinkage toward 0.5.

| $\eta$ | GSM8K @ 800 | | HotpotQA @ 1000 | |
|---|---|---|---|---|
| | AVA | Always-Verify | AVA | Always-Verify |
| 0.0 | **82.1%** | 81.7% | **68.3%** | 64.2% |
| 0.1 | **81.6%** | 81.1% | **67.1%** | 62.9% |
| 0.2 | **80.9%** | 80.3% | **65.8%** | 61.2% |
| 0.3 | **79.8%** | 79.0% | **63.9%** | 59.4% |

Table 13: Verification cascade efficiency. Most examples exit early at lightweight verification levels, achieving significant cost savings compared to uniform full verification.

| Dataset | Level 1 Exit | Level 2 Exit | Level 3 Full | Cost Savings |
|---|---|---|---|---|
| GSM8K | 72% | 21% | 7% | 64% |
| HotpotQA | 65% | 23% | 12% | 58% |
| HumanEval | 68% | 20% | 12% | 61% |
| Average | **68%** | **22%** | **10%** | **62%** |

