# OpenReview forum: "Anytime Verified Agents: Adaptive Compute Allocation for Reliable LLM Reasoning under Budget Constraints"
_TMLR — Accepted by TMLR_

### Review · Reviewer_uHXr · 2025-12-18

**Summary Of Contributions:**

This paper proposes a rule-based dynamic budget allocation scheme for test-time compute in LLMs. It relies on four aspects of uncertainty of the LLM prediction to distribute a user-specified overall budget to number of new samples generated, depth and branching factors for adaptive search, verifier effort (abstracted to four levels) and whether to call tools. While the both the confidence estimation and allocation rules are simplistic so far, it is undeniable that the concept holds tremendous potential for test-time compute efficiency and the experiments show that even this simple approach can meaningfully improve performance on low budgets.

**Audience:**

Yes

**Audience Explanation:**

Test-time compute optimization is incredibly relevant with the recent rise of methods relying on it. Thus I believe this paper is very timely.

**Broader Impact Concerns:**

No specifics.

**Claims And Evidence:**

Yes

**Claims Explanation:**

The evaluation shows that the proposed allocation scheme increases efficiency on the tested datasets and the additional results on ablating the different components and the verification efficiency support these findings.

In my opinion, the main points of criticism of the evaluation lie in the scale of the experiments (single model, benchmark selection). I hesitate to make my score dependent on these factors, however, since the experiments are certainly challenging to run. I also believe that the current scope already gives a strong signal on the performance of the allocation rules and especially the confidence estimation.

The other point would be code availability. I believe open code is important for open science and thus think the authors should release their experimental setup at the latest upon acceptance.

**Requested Changes:**

- I appreciate Figure 1 as an overview and since it already takes up a full page, I would suggest increasing font size everywhere to improve its readability.
- The discussion sections seems like a list of keywords at the moment, several of which would warrant at least one or two sentences of full discussion of their own. I think this should be improved.
- Several Tables are not referenced in the text which hurts the readability of the results section overall, this should be corrected.
- There is currently an empty "Acknowledgements" Section above the references. If you don't plan on using it upon publication, this should be removed.

---

> ### Comment · Reviewer_uHXr · 2026-01-06
> **Paper Revision**
>
> The updates, especially the discussion section, improve the paper and implement the changes requested in my review.

---

### Review · Reviewer_X5NB · 2026-01-17

**Summary Of Contributions:**

This paper proposes a framework designed to dynamically allocate computational resources for LLM agents under a unified budget.

**Audience:**

Yes

**Audience Explanation:**

LLM agent is a popular research topic nowadays.

**Claims And Evidence:**

Yes

**Claims Explanation:**

The paper addresses a practical and significant problem: managing the trade-off between inference cost and reliability across multiple dimensions (search, sampling, verification) simultaneously.

The reported results demonstrate clear Pareto dominance over the chosen baselines.

**Requested Changes:**

1. Robustness of Heuristics: How were the specific thresholds (e.g., for 10 samples) in the controller derived? Did you perform a sensitivity analysis? I would like to see a plot showing Cost vs. Accuracy as these thresholds are perturbed.

2. Generalization of Calibration: The paper mentions that the calibration is performed per dataset. How does the system perform if the calibration mapping is transferred? For example, training the calibrator on GSM8K and testing on MATH (or a harder subset). Does the rule-based controller fail if the uncertainty estimates become uncalibrated?

3. Please discuss the main difference of AVA with the following works.

[1]. SelfBudgeter: Adaptive Token Allocation for Efficient LLM Reasoning
[2]. Strategic Scaling of Test-Time Compute: A Bandit Learning Approach

---

### Review · Reviewer_xo3r · 2026-02-04

**Summary Of Contributions:**

The paper proposes Anytime Verified Agents (AVA), a framework for adaptive compute allocation in LLM agents that dynamically routes budget across sampling, search, verification, and tool use. AVA combines a calibrated uncertainty estimator, a multi-level verification cascade with early exits, and a value-of-information–guided search, coordinated by a budget-aware controller to either meet a target reliability at minimal cost or maximize reliability under a fixed budget. Experiments on GSM8K, HotpotQA, and HumanEval report 20–40% cost reductions at comparable reliability versus static baselines (fixed-depth search, fixed-sample self-consistency, and always-verify), with ablations suggesting each component contributes to efficiency.

**Additional Comments:**

First, I would like to apologize for the delay in submitting this review. I hope that the detailed and candid feedback provided here compensates for the wait. In general, I believe the paper requires major revisions, particularly from a presentation perspective. It would benefit significantly from greater mathematical rigor and formality. I suggest dedicating time to refining these aspects and ensuring that the core concepts of the framework, which touches on many facets of LLM operations, are defined explicitly rather than given for granted.

**Audience:**

Yes

**Audience Explanation:**

Yes, I think that the paper may be relevant for TMLR's audience since it tackle a actual problem related to the computational cost of LLMs.

**Broader Impact Concerns:**

The authors do not present a specific Broader Impact Statement. I do not see immediate critical ethical concerns requiring a halt to publication.

**Claims And Evidence:**

No

**Claims Explanation:**

The claims made in the manuscript are mainly supported by a relevant experimental campaign, which empirically demonstrates the effectiveness of the method in three different tasks.

However, the work suffers significantly from poor presentation. In particular, it lacks formality in variable definitions, naming conventions, and rigorousness. Specific issues include:
- Several values are not formally defined, such as task complexity $t$, $d_{max}$, $n_{max}$, $f$ function in the isotonic regression, $v_l$ output from verifiers, $expand(n)$, $g(n)$, $c_{expand}$. Even if their meaning can be inferred, a higher degree of formality is expected.
- Several thresholds are set to specific values without explanation or intuition (e.g., $\tau_{entropy}$, adaptive branching thresholds, default $r_{target}=0.9$, $\Delta C$).
- Notation inconsistency: remaining budget oscillates between $B_r$ and $b_r$; confidence $c$ is later referred to as $confidence_{raw}$ (why do not use $c$ with a proper subscript?); the controller's state space is defined first without the history dimension $h_t$, and then later with it.
- The if-then-else conditions for sampling, search, and verification are written such that both the first and second conditions could be true simultaneously. This should be presented as a formal system of equations or via pseudocode to avoid ambiguity.
- In general, I advise against defining variables with full words; symbols with subscripts are preferable. Furthermore, specific threshold values are currently embedded in the methodology text. It would be better to use symbols in the equations and report the specific values and justifications later in the paper or in an appendix.

Furthermore, these threshold values seem specific to these applications and the model used (GPT-5). The general adaptiveness of the method is not immediately evident (since it is based on rule-based strategies and heuristics), although the method is indeed interpretable.

**Requested Changes:**

- The paper needs to be more clear: I suggest to revise the paper organization. For example, you can leverage the 3 stages structure to present sections 3.2, 3.3, 3.4 and 3.5 as subsections under the relative stage. This would improve redability and clarity; I sometimes struggled to understand where I was in the flow of the algorithm. The scheme in Figure 1 is instead very helpful, you should exploit it more.
- The paper requires more formality. Please refer to weaknesses highlighted before about threshold and undefined variables, and ensure the method is mathematically rigorous.
- Please add the missing citation for PAVA.
- Please make the paper self-contained. Some concepts are taken for granted. Since TMLR does not have strict page limits, you can expand on these (e.g., explicitly describe what a "verifier" is, provide a dedicated section describing baselines, explain what "tool calls" entail, and detail how the $r(B)$ function is decided).
- How do you compute the uncertainty $u$ from current state? I think this is not fully explained.
- Please explicitely report fixed random seeds for reproducibility.

---

### Review · Reviewer_2VfJ · 2026-02-14

**Summary Of Contributions:**

This paper introduces Anytime Verified Agents (AVA), a framework that adaptively allocates test-time compute (sampling, search, verification, tool use) to maximize reliability under user-specified budgets. A budget-aware controller makes iterative allocation decisions from a state including calibrated confidence and remaining budget, guided by value-of-information principles. Uncertainty is aggregated from multiple signals and calibrated via post-hoc isotonic regression, enabling early stopping and selective escalation through a verification cascade and VoI-guided search.

**Audience:**

Yes

**Audience Explanation:**

The paper will interest readers working on LLM inference, evaluation, reliability, and ML systems. Its budget-aware framework for adaptive test-time compute allocation (sampling/search/verification/tool use) addresses a practical deployment problem—improving reliability under constrained cost—and provides actionable design ideas (calibration, verification cascades, early stopping) relevant to both methodology and systems communities.

**Broader Impact Concerns:**

The work is primarily a systems/methodology contribution for improving reliability under cost constraints, so direct ethical risk is low.

**Claims And Evidence:**

Yes

**Claims Explanation:**

The submission provides convincing evidence for its primary empirical claim: AVA effectively improves the cost–quality trade-off via adaptive compute allocation, as demonstrated on GSM8K, HotpotQA, and HumanEval.

However, the evidence for its broader "anytime" robustness is less complete. The framework relies heavily on specific verifier setups and calibrated uncertainty, yet the evaluation lacks sensitivity analyses regarding verifier quality or distribution shifts (where calibration typically degrades). Thus, while the gains are well-supported in the reported settings, the system's stability under varying component quality remains to be fully established.

**Requested Changes:**

Critical:
1. The evaluation primarily compares AVA to static allocation baselines (fixed-depth search, fixed-N self-consistency, always-verify). To justify the added complexity of the MDP-style controller and VoI components, the paper should include comparisons to simpler adaptive baselines, e.g., a confidence-threshold early-exit policy or a lightweight adaptive sampling rule, which are conceptually close to prior “confidence-based halting/early stopping” ideas discussed in related work. If these heuristics achieve similar savings, the necessity of AVA’s heavier machinery would be less clear.

Strengthening：
1. AVA relies on calibrated uncertainty and verifier reliability—both can fail under domain shift or verifier noise. The paper should add OOD transfer evaluations and stress tests with degraded/noisy verifiers to demonstrate that the reported cost–quality frontier remains stable under imperfect conditions.

---

### Decision · Action_Editor_X29p · 2026-04-08

**Recommendation:** Accept with minor revision

**Audience:**

Yes

**Audience Explanation:**

The paper should be of sufficient interest for the LLM reasoning community.

**Claims And Evidence:**

Yes

**Claims Explanation:**

I think the paper presents interesting results regarding budget-constrained reasoning, which is a timely topic given the prevalence of test-time scaling nowadays. While the proposed method is a heuristic, it seems to be effective and is supported by sufficient empirical results. I will suggest the authors to include the additional results in the rebuttal and if possible, also to include results with more challenging benchmarks (GSM8K is pretty outdated today). Overall, I recomment to accept the paper given the minor revision meets the standard.